# Pathway-, layer- and cell-type-specific thalamic input to mouse barrel cortex

B Semihcan Sermet[1†], Pavel Truschow[2], Michael Feyerabend[2], Johannes M Mayrhofer[1], Tess B Oram[3], Ofer Yizhar[3], Jochen F Staiger[2], Carl CH Petersen[1]*

[1]Laboratory of Sensory Processing, Brain Mind Institute, Faculty of Life Sciences, Ecole Polytechnique Fédérale de Lausanne (EPFL), Lausanne, Switzerland; [2]Institute for Neuroanatomy,University Medical Center, Georg-August-University Goettingen, Goettingen, Germany; [3]Department of Neurobiology, Weizmann Institute of Science, Rehovot, Israel

**Abstract** Mouse primary somatosensory barrel cortex (wS1) processes whisker sensory information, receiving input from two distinct thalamic nuclei. The first-order ventral posterior medial (VPM) somatosensory thalamic nucleus most densely innervates layer 4 (L4) barrels, whereas the higher-order posterior thalamic nucleus (medial part, POm) most densely innervates L1 and L5A. We optogenetically stimulated VPM or POm axons, and recorded evoked excitatory postsynaptic potentials (EPSPs) in different cell-types across cortical layers in wS1. We found that excitatory neurons and parvalbumin-expressing inhibitory neurons received the largest EPSPs, dominated by VPM input to L4 and POm input to L5A. In contrast, somatostatin-expressing inhibitory neurons received very little input from either pathway in any layer. Vasoactive intestinal peptide-expressing inhibitory neurons received an intermediate level of excitatory input with less apparent layer-specificity. Our data help understand how wS1 neocortical microcircuits might process and integrate sensory and higher-order inputs.

*For correspondence:
carl.petersen@epfl.ch

Present address: †Sorbonne Université, INSERM, CNRS, Institut de la Vision, Paris, France

Competing interests: The authors declare that no competing interests exist.

## Introduction

Thalamic input is of critical importance for neocortical function, but the distribution of synaptic input onto distinct cell-types located in different cortical layers remains incompletely understood. All cortical areas receive thalamic input, with different thalamic nuclei projecting to different cortical regions and layers (*Clascá et al., 2012*; *Herkenham, 1980*). Primary sensory cortical areas receive thalamic input from primary sensory thalamic nuclei - for example the lateral geniculate nucleus innervates primary visual cortex, the medial geniculate nucleus innervates primary auditory cortex and the lateral and medial portions of the ventral posterior nucleus innervate primary somatosensory cortex. The primary sensory thalamic input to primary sensory areas is arranged in topographic maps, for example retinotopy in visual cortex, tonotopy in auditory cortex, and somatotopy in somatosensory cortex. Anatomically, the innervation from primary sensory thalamus is densest in layer 4 (L4) of the neocortex, but also present in all other cortical layers (*Oberlaender et al., 2012a*). The primary sensory thalamic input is thought to drive sensory processing in the recipient cortex (*Reinhold et al., 2015*), and it is therefore of critical importance to investigate thalamic input onto specific classes of neurons distributed across cortical layers.

Mice receive important tactile sensory information about their immediate surroundings from their array of mystacial whiskers (*Diamond et al., 2008*; *Feldmeyer et al., 2013*; *Petersen, 2019*). The whisker representation in mouse primary somatosensory cortex (wS1) is arranged in discrete anatomical units, termed 'barrels', with each barrel representing an individual whisker on the snout (*Woolsey and Van der Loos, 1970*). The barrels are found in L4 of wS1, and each barrel is densely

innervated by neurons located in the homologous 'barreloid' of the ventral posterior medial (VPM) primary whisker somatosensory thalamus (*Arnold et al., 2001*; *Oberlaender et al., 2012a*). Although VPM axons are densest in L4, there is also substantial VPM axon innervation of L3 above the barrels and at the border of L5 and L6 (*Wimmer et al., 2010*). In many cases, the dendritic arborizations of postsynaptic neurons in wS1 span different cortical layers and therefore neurons with cell-bodies located in many different, if not all, layers could receive synaptic input from VPM (*Meyer et al., 2010*; *Oberlaender et al., 2012a*). Consistent with this hypothesis, previous electrophysiological studies have found that stimulation of VPM axons evokes excitatory postsynaptic potentials (EPSPs) in various cortical layers (*Bureau et al., 2006*; *Constantinople and Bruno, 2013*; *Cruikshank et al., 2010*; *Petreanu et al., 2009*). There are many different types of neocortical neurons, with largely non-overlapping populations of inhibitory GABAergic neurons being classified through expression of parvalbumin (PV), somatostatin (SST) and vasoactive intestinal peptide (VIP) (*Feldmeyer et al., 2018*; *Lee et al., 2010*; *Tasic et al., 2018*). Previous work has found that VPM provides the largest synaptic input to PV neurons (*Cruikshank et al., 2007*), and substantially less to SST neurons, but systematic quantification of VPM input to different cell-types across layers in a single study is currently lacking. To fill these gaps in knowledge, here we investigate the relative strength of excitatory synaptic input from VPM to different cell-types across layers in wS1, helping to understand how these neurons process tactile sensorimotor information (*Gentet et al., 2012*; *Gentet et al., 2010*; *Lee et al., 2013*; *Muñoz et al., 2017*; *Yu et al., 2019*; *Yu et al., 2016*).

Importantly, wS1 is also strongly innervated by the medial part of the posterior (POm) nucleus of the thalamus. POm is considered a higher-order thalamic area, since it receives strong 'driver' input from excitatory L5B neurons in wS1 (*Groh et al., 2008*; *Mease et al., 2016a*), and inactivation of wS1 potently inhibits both spontaneous and whisker deflection-evoked action potential firing of many POm neurons (*Diamond et al., 1992*; *Mease et al., 2016b*). POm axons densely innervate L1 and L5A in wS1, forming a complementary pattern to the VPM innervation (*Wimmer et al., 2010*). Stimulation of POm axons evokes EPSPs in excitatory neurons with cell bodies in various layers, with strong responses found in L5A (*Audette et al., 2018*; *Bureau et al., 2006*; *Petreanu et al., 2009*). Large responses to POm stimulation were also found in PV neurons (*Audette et al., 2018*). Building upon this knowledge, here, we also systematically quantify the relative strength of excitatory input from POm onto different cell-types across layers of wS1.

## Results

### Optogenetic stimulation of VPM and POm input to wS1

We used adeno-associated viruses to express channelrhodopsin-2 (ChR2) in either VPM or POm thalamus. In some experiments (n = 12 mice), we used GPR26-Cre mice, which specifically express Cre-recombinase in higher-order thalamus (including POm) but not first-order thalamus (i.e. VPM) (*Gerfen et al., 2013*). Injection of virus encoding Cre-ON sequences (FLEX/DIO) into these mice, induced expression in POm, but not VPM. Conversely, injection of virus encoding a Cre-OFF element (DFO) (*Fenno et al., 2014*) induced expression in VPM, but not POm. Consistent with previously published anatomy in rats (*Meyer et al., 2010*; *Wimmer et al., 2010*), we found that in mice VPM strongly innervated L4 barrels in wS1, as well as innervating L3 above the barrels and the border of L5 and L6, whereas POm strongly innervated L5A and L1 (*Figure 1A*). In other experiments (n = 64 mice), we stereotactically injected virus without Cre-dependence into VPM or POm, finding similar innervation patterns to those observed in the GPR26-Cre mice. In this study, we therefore combined the data from Cre-dependent and Cre-independent expression of ChR2.

In order to study thalamocortical input to wS1, we obtained whole-cell membrane potential recordings from wS1 neurons in parasagittal brain slices of mice aged P42-92 containing thalamic axons expressing ChR2, and applied 1 ms blue light flashes to evoke neurotransmitter release. We pharmacologically blocked fast GABAergic synaptic transmission by applying picrotoxin (50 µM) to the extracellular solution. To prevent polysynaptic activity, we further added TTX (1 µM) and 4-AP (100 µM) (*Petreanu et al., 2009*). Under these experimental conditions, we found that blue light flashes evoked monosynaptic EPSPs in wS1 neurons, which were completely blocked by application of CNQX (20 µM) and APV (50 µM) to block AMPA and NMDA ionotropic glutamate receptors (*Figure 1B*).

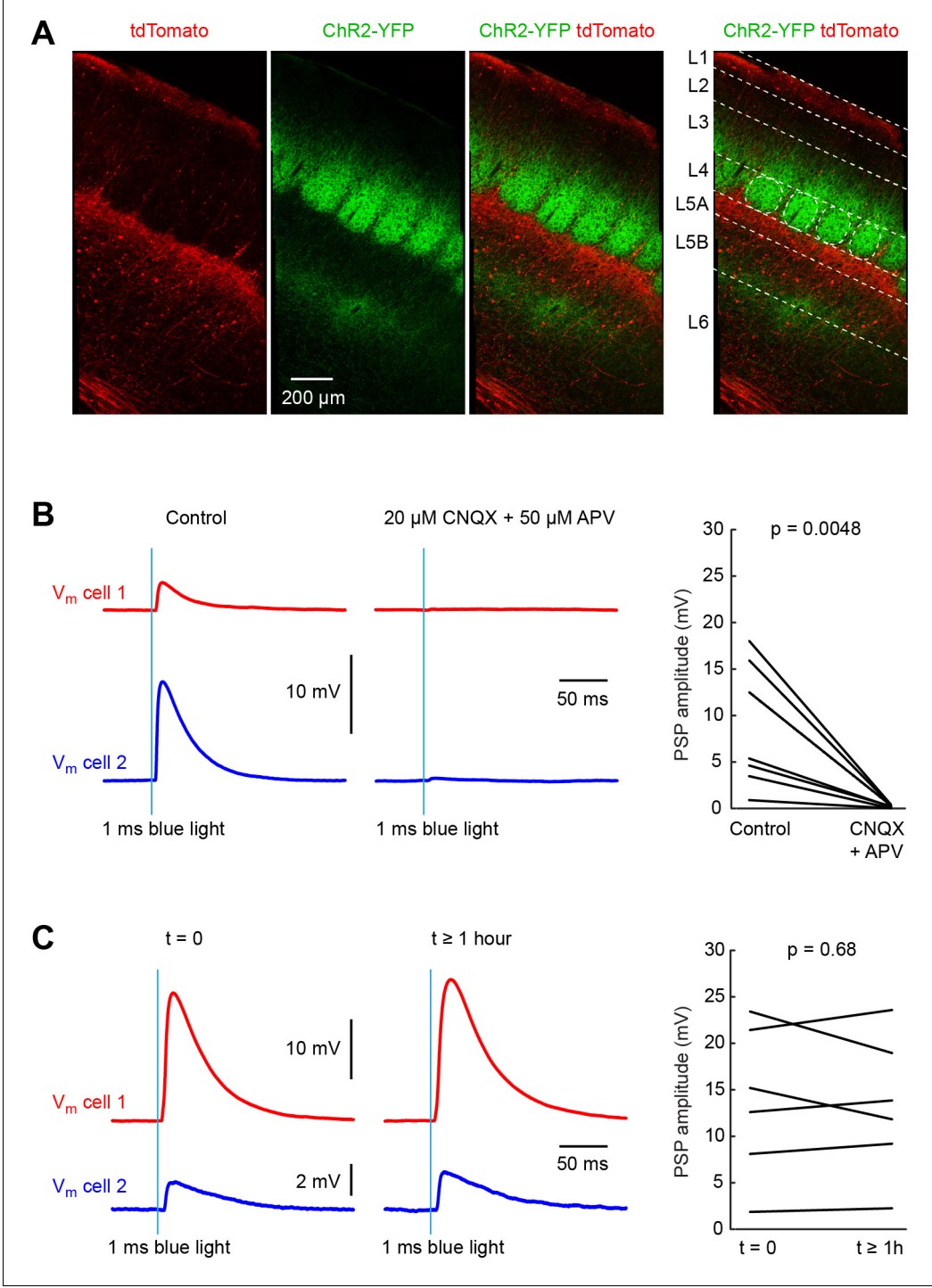

**Figure 1.** Pathway-specific optogenetic stimulation of thalamus evokes stable monosynaptic input to neurons in whisker primary somatosensory barrel cortex (wS1). (**A**) A Cre-ON (FLEX) AAV virus expressing tdTomato and a Cre-OFF (DFO) virus expressing YFP, were injected together into the thalamus of GPR26-Cre mice, which express Cre in the higher-order somatosensory thalamic nucleus POm, but not in the first order somatosensory thalamic nucleus VPM. Thalamic axons from POm (red) were prominent in L5A and L1 in wS1. Axons from VPM (green) aggregate in clusters termed barrels in L4. Additional VPM innervation is found at the L5/L6 border. (**B**) Example average traces of membrane potential ($V_m$) from two simultaneously-recorded excitatory L4 neurons during optogenetic stimulation of thalamic VPM axons in wS1 (left). Application of CNQX (AMPA receptor antagonist) and APV (NMDA receptor anatagonist) completely blocked the evoked EPSP in the example experiment (middle),

*Figure 1 continued on next page*

*Figure 1 continued*

and on average across 8 cells (right) (p=0.0048, two-tailed Wilcoxon signed-rank test). (**C**) Example average traces of membrane potential (V$_m$) from two simultaneously-recorded excitatory L4 neurons in wS1 during optogenetic stimulation of thalamic VPM axons shortly after establishment of the whole-cell recordings (left) and more than an hour later (middle). On average across 6 cells, there was no systematic change in EPSP amplitude over the ~1 hr period of the experiments (right) (p=0.68, two-tailed Wilcoxon signed-rank test).

The online version of this article includes the following source data for figure 1:

**Source data 1.** Data values and statistics underlying *Figure 1*.

To assess the thalamic input to different cells across cortical layers, we aimed to record from as many neurons per slice as possible, in order to be able to make direct within slice comparisons. Typically, we recorded from two cells simultaneously and then quickly moved on to record from further neurons in the same slice using the same optical stimulation parameters. The total duration of recordings from the same slice could last for over an hour, and it was therefore of critical importance that the optogenetic stimulation was consistent over this time period. In control experiments, we therefore made long-lasting recordings for over an hour and measured optogenetically-evoked EPSPs over time. We found that on average there was no change in the amplitude of evoked EPSPs comparing those recorded in the first minutes of the whole-cell recording to those measured more than an hour later in the same cells (*Figure 1C*).

## VPM input to excitatory neurons across layers in wS1

Having established methods for reliably recording monosynaptic thalamic input to wS1, we next recorded from excitatory neurons across layers in mice expressing ChR2 in VPM (*Figure 2A*). Neurons were filled with biocytin through the whole-cell recording electrode, and after fixation were fluorescently stained with streptavidin conjugated to Alexa 647. The neuronal morphology could therefore be visualized with respect to the thalamic axonal innervation. The dendritic trees of a subset of recorded neurons were digitally reconstructed from confocal stacks. In the example experiment (*Figure 2A*), as well as in general, the largest PSPs were found in L4 neurons.

Across recordings from different slices (n = 33 slices, n = 33 mice), we found clear differences in VPM input to neurons in different cortical layers (*Figure 2B*). The mean peak amplitude of EPSPs according to the laminar location of the cell somata was: L2, 2.0 ± 5.3 mV (n = 29 cells); L3, 6.7 ± 7.2 mV (n = 45 cells); L4, 14.5 ± 7.4 mV (n = 65 cells); L5A, 1.7 ± 2.1 mV (n = 39 cells); L5B, 2.3 ± 2.9 mV (n = 44 cells); and L6, 3.5 ± 6.7 mV (n = 40 cells). Because of differing levels of ChR2 expression across slices, we normalized the EPSP amplitudes to the mean response of the cells recorded in L4 within each slice. The normalized peak EPSP amplitudes across layers were: L2, 0.20 ± 0.54 (n = 29 cells); L3, 0.46 ± 0.50 (n = 45 cells); L4, 1 ± 0.37 (n = 65 cells); L5A, 0.14 ± 0.21 (n = 39 cells); L5B, 0.17 ± 0.22 (n = 44 cells); and L6, 0.39 ± 0.79 (n = 40 cells) (*Figure 2B* and *Table 1*). Excitatory neurons across all cortical layers therefore receive input from VPM, and in each layer some neurons receive much stronger input than the average.

The early slope of EPSPs relates more directly to synaptic currents compared to the peak amplitude of the EPSPs. We therefore also computed the slope of the rising phase of the EPSP, finding similar layer differences to those found for the EPSP amplitudes. The mean EPSP slope across layers was: L2, 0.86 ± 2.3 mV/ms (n = 29 cells); L3, 2.9 ± 3.3 mV/ms (n = 45 cells); L4, 5.3 ± 3.5 mV/ms (n = 65 cells); L5A, 0.56 ± 0.74 mV/ms (n = 39 cells); L5B, 0.70 ± 0.89 mV/ms (n = 43 cells); and L6, 1.2 ± 2.7 mV/ms (n = 39 cells). To better compare across slices, we again normalized the EPSP slope to the mean of the slope of the EPSPs in L4 neurons for each slice. The normalized EPSP slope across layers was: L2, 0.19 ± 0.50 (n = 29 cells); L3, 0.51 ± 0.54 (n = 45 cells); L4, 1 ± 0.43 (n = 65 cells); L5A, 0.11 ± 0.16 (n = 39 cells); L5B, 0.16 ± 0.23 (n = 43 cells); and L6, 0.35 ± 0.81 (n = 39 cells) (*Figure 2B* and *Table 2*).

For both EPSP amplitude and slope, VPM input appears to be roughly twice as large in L4 compared to L3, and three times as large in L4 compared to L6. On average, inputs to other layers (L2, L5A and L5B) were smaller, although it is important to note that in each layer we found neurons with large responses. The amplitude and slope of both normalized and absolute EPSPs in L4 were significantly larger than the input to any other layer (two-tailed Wilcoxon rank-sum test, p<10$^{-7}$ for each comparison).

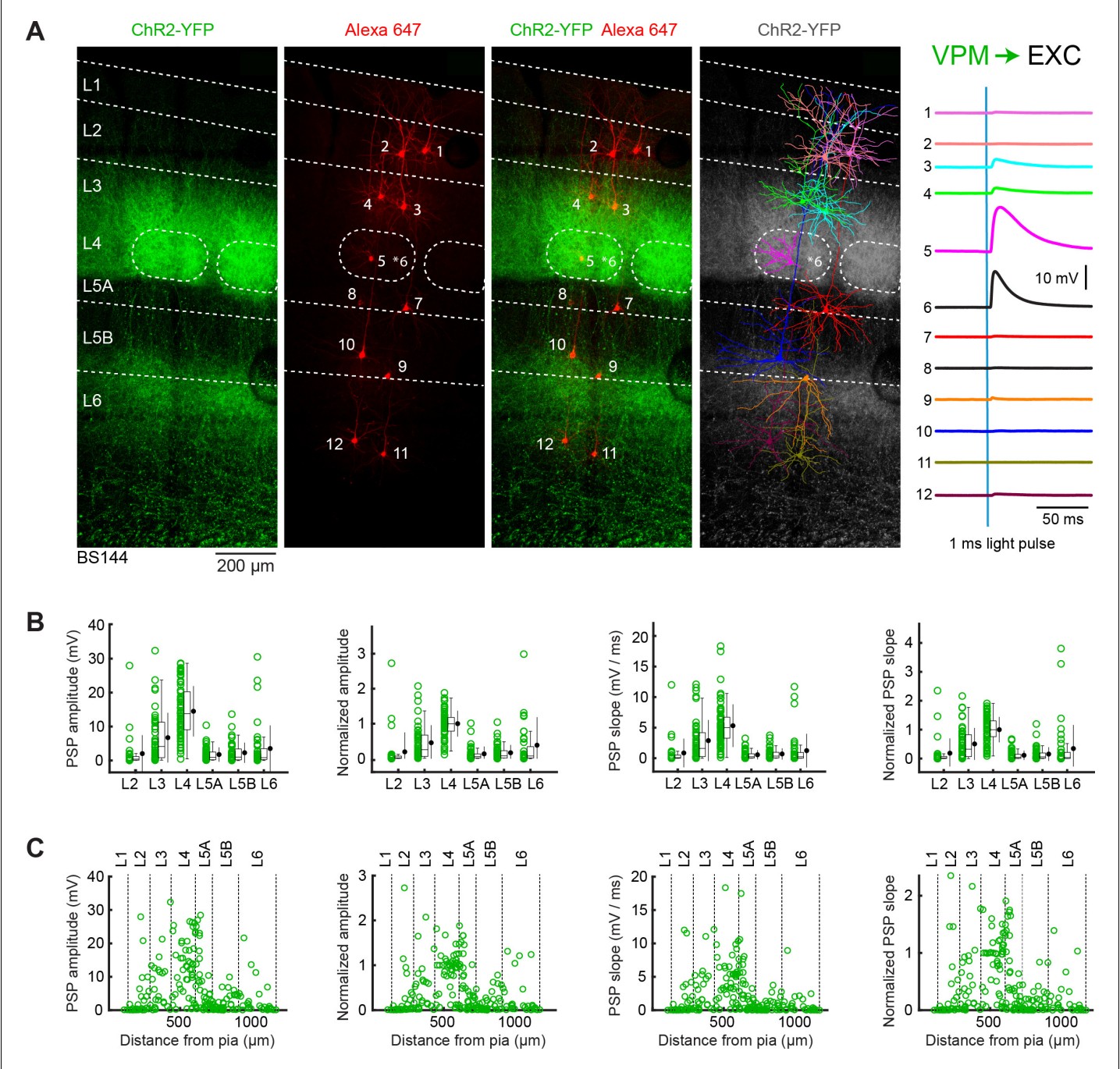

**Figure 2.** VPM input to excitatory neurons across layers in wS1. (**A**) An example experiment (BS144) in which ChR2 was expressed in VPM and whole-cell recordings were obtained sequentially from 12 excitatory neurons across different cortical layers in wS1. Neurons were filled with biocytin and *post hoc* stained with streptavidin conjugated to Alexa 647 to reveal dendritic morphology, which was digitally reconstructed. Neurons #5 and #6 (both located in L4, although the full morphology of #6 was not recovered, its location is indicated by an asterisk) received the largest EPSPs in response to optogenetic stimulation of VPM. (**B**) The laminar location was assigned for each recording. The peak amplitude of the evoked EPSP averaged across trials was determined for each cell, and plotted for each layer along with the median (box plot, including interquartile range and whiskers) and the mean (filled circle, along with SD error bars) (left). The EPSP amplitudes from individual slices were normalized to the mean amplitude of the EPSP measured in the L4 neurons of the same slice (center left). The slope of the rising phase of the EPSP was determined and plotted across layers (center right). The normalized EPSP slope was calculated by dividing by the mean slope of the EPSPs in the L4 neurons recorded in the same slice (right). (**C**) EPSP amplitude (left), normalized EPSP amplitude (center left), EPSP slope (center right) and normalized EPSP slope (right) plotted as a function of cortical depth below the pial surface. Approximate layer boundaries are drawn according to *Lefort et al. (2009)*.

The online version of this article includes the following source data for figure 2:

**Source data 1.** Data values underlying *Figure 2*.

**Table 1.** Normalized EPSP amplitudes of VPM input including: mean ± SD, *median*, and n = number of recorded cells.

| VPM amplitude | EXC | PV | SST | VIP |
|---|---|---|---|---|
| L2 | 0.20 ± 0.54 *0.022* n = 29 | 0.04 ± 0.02 *0.032* n = 3 | 0.01 ± 0.00 *0.0082* n = 3 | 0.13 ± 0.26 *0.024* n = 10 |
| L3 | 0.46 ± 0.50 *0.26* n = 45 | 0.35 ± 0.34 *0.19* n = 9 | 0.09 ± 0.10 *0.025* n = 3 | 0.32 ± 0.49 *0.046* n = 17 |
| L4 | 1.00 ± 0.37 *1.00* n = 65 | 1.15 ± 0.92 *1.05* n = 14 | 0.09 ± 0.08 *0.067* n = 8 | 0.15 ± 0.16 *0.089* n = 8 |
| L5A | 0.14 ± 0.21 *0.063* n = 39 | 0.12 ± 0.22 *0.016* n = 5 | 0.03 ± 0.01 *0.021* n = 6 | 0.03 ± 0.01 *0.029* n = 5 |
| L5B | 0.17 ± 0.22 *0.080* n = 44 | 0.04 ± 0.02 *0.045* n = 11 | 0.01 ± 0.01 *0.0073* n = 9 | 0.13 ± 0.23 *0.028* n = 5 |
| L6 | 0.39 ± 0.79 *0.077* n = 40 | 0.17 ± 0.20 *0.033* n = 5 | 0.01 ± 0.01 *0.0060* n = 4 | *0.011* n = 1 |

The assignment of neurons to individual layers was based upon multiple qualitative features characterized during the recording and in the anatomical DAPI-stained slices including: thalamic innervation, cell density, cell soma shape and size, and the prominence of apical dendrites. The boundaries between layers are not always easy to define. We therefore also plotted our data with respect to subpial depth (*Figure 2C*), finding that our results are consistent with the largest inputs arriving in L4 and the overlying L3.

## POm input to excitatory neurons across layers in wS1

In another set of mice (n = 43 slices, n = 43 mice), we expressed ChR2 in POm, and again recorded neurons across layers to examine the laminar distribution of higher-order thalamic input to wS1 (*Figure 3A*). Consistent with the strong axonal innervation of L5A, we found that the largest EPSPs were typically found in L5A neurons, as shown in the example experiment (*Figure 3A*). The mean

**Table 2.** Normalized EPSP slopes of VPM input including: mean ± SD, *median*, and n = number of recorded cells.

| VPM slope | EXC | PV | SST | VIP |
|---|---|---|---|---|
| L2 | 0.19 ± 0.50 *0.018* n = 29 | 0.03 ± 0.03 *0.021* n = 3 | 0.02 ± 0.01 *0.016* n = 3 | 0.19 ± 0.35 *0.021* n = 7 |
| L3 | 0.51 ± 0.54 *0.33* n = 45 | 0.85 ± 1.07 *0.38* n = 9 | 0.03 ± 0.05 *0.0049* n = 3 | 0.40 ± 0.61 *0.038* n = 16 |
| L4 | 1.00 ± 0.43 *1.00* n = 65 | 2.73 ± 2.24 *1.68* n = 14 | 0.06 ± 0.04 *0.060* n = 8 | 0.09 ± 0.08 *0.044* n = 8 |
| L5A | 0.11 ± 0.16 *0.043* n = 39 | 0.38 ± 0.76 *0.0079* n = 5 | 0.01 ± 0.01 *0.0061* n = 6 | 0.02 ± 0.02 *0.0057* n = 5 |
| L5B | 0.16 ± 0.23 *0.060* n = 43 | 0.06 ± 0.04 *0.077* n = 11 | 0.00 ± 0.00 *0.0016* n = 6 | 0.16 ± 0.24 *0.024* n = 4 |
| L6 | 0.35 ± 0.81 *0.037* n = 39 | 0.13 ± 0.16 *0.015* n = 5 | 0.06 ± 0.05 *0.082* n = 3 | *0.00072* n = 1 |

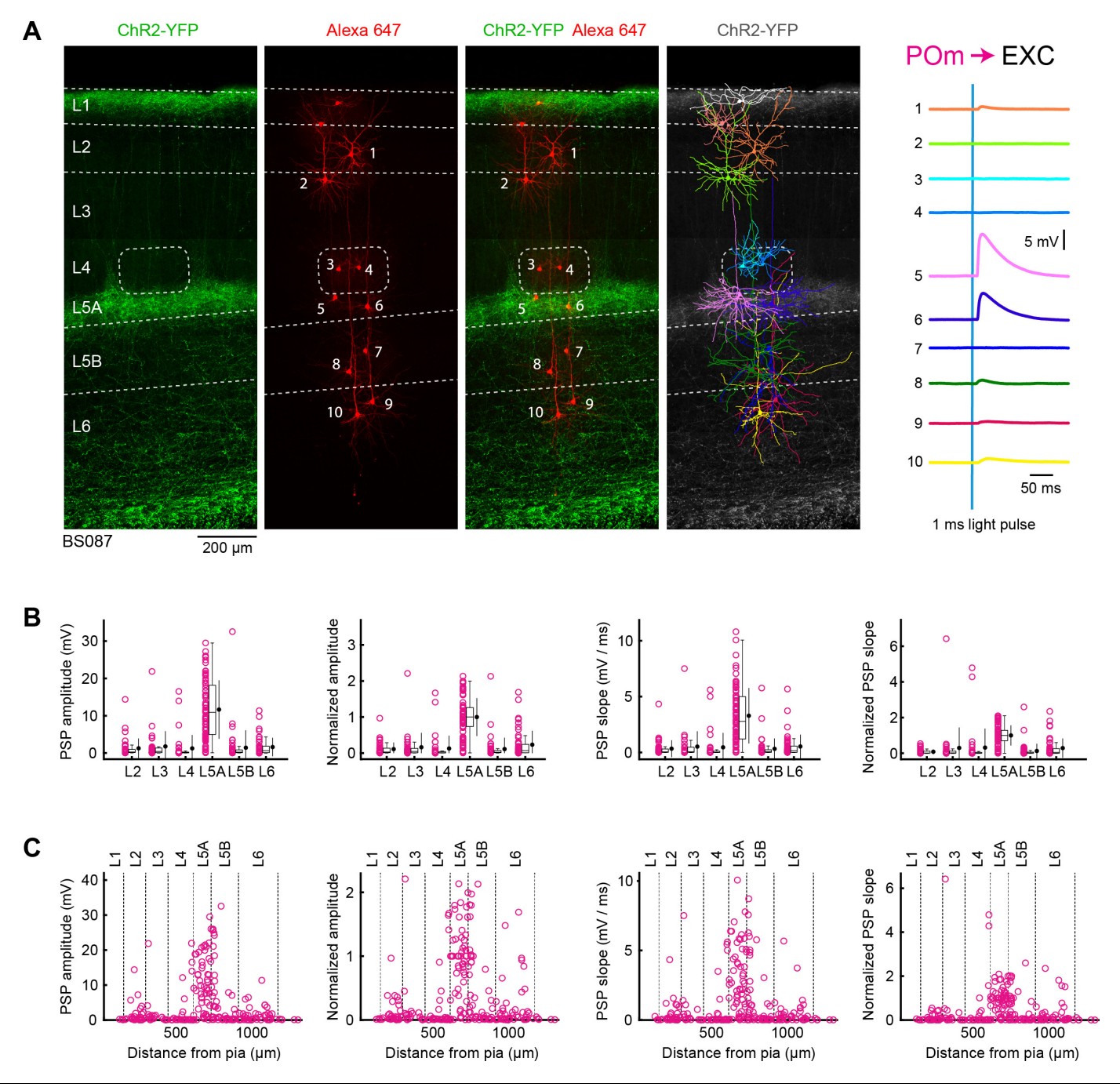

**Figure 3.** POm input to excitatory neurons across layers in wS1. (**A**) ChR2 was expressed in POm in the example experiment (BS087) and 10 excitatory neurons across different cortical layers in wS1 were recorded sequentially. Neurons were filled with biocytin and *post hoc* stained with streptavidin conjugated to Alexa 647 to reveal dendritic morphology, which was digitally reconstructed. Neurons #5 and #6 (both located in L5A) received the largest EPSPs in response to optogenetic stimulation of POm. (**B**) The location of the soma of each recorded neuron was assigned to a cortical layer, and the peak amplitude of the evoked EPSP averaged across trials was determined for each cell, and plotted for each layer along with the median (box plot, including interquartile range and whiskers) and the mean (filled circle, along with SD error bars) (left). The EPSP amplitudes from individual slices were normalized to the mean amplitude of the EPSP measured in the L5A neurons of the same slice (center left). The slope of the rising phase of the EPSP was determined and plotted across layers (center right). The normalized EPSP slope was calculated by dividing by the mean slope of the EPSPs in the L5A neurons recorded in the same slice (right). (**C**) EPSP amplitude (left), normalized EPSP amplitude (center left), EPSP slope (center right) and normalized EPSP slope (right) plotted as a function of cortical depth below the pial surface. Approximate layer boundaries are drawn according to *Lefort et al. (2009)*.

*Figure 3 continued on next page*

*Figure 3 continued*

The online version of this article includes the following source data for figure 3:

**Source data 1.** Data values underlying *Figure 3*.

peak amplitude of EPSPs for each layer was: L2, 1.3 ± 2.6 mV (n = 38 cells); L3, 1.8 ± 4.0 mV (n = 31 cells); L4, 1.3 ± 3.5 mV (n = 38 cells); L5A, 11.7 ± 7.8 mV (n = 75 cells); L5B, 1.4 ± 4.6 mV (n = 52 cells); and L6, 1.6 ± 2.4 mV (n = 47 cells) (*Figure 3B*). Normalizing the peak EPSP amplitude to the mean measured in L5A within each slice, we found: L2, 0.11 ± 0.18 (n = 38 cells); L3, 0.16 ± 0.39 (n = 31 cells); L4, 0.13 ± 0.35 (n = 38 cells); L5A, 1 ± 0.52 (n = 75 cells); L5B, 0.11 ± 0.31 (n = 52 cells); and L6, 0.23 ± 0.38 (n = 47 cells) (*Figure 3B* and *Table 3*). The mean EPSP slope across layers was: L2, 0.33 ± 0.74 mV/ms (n = 38 cells); L3, 0.52 ± 1.35 mV/ms (n = 31 cells); L4, 0.46 ± 1.27 mV/ms (n = 35 cells); L5A, 3.28 ± 2.46 mV/ms (n = 75 cells); L5B, 0.32 ± 0.91 mV/ms (n = 50 cells); and L6, 0.53 ± 1.05 mV/ms (n = 44 cells). The EPSP slope normalized to that of L5A was: L2, 0.08 ± 0.13 (n = 38 cells); L3, 0.30 ± 1.13 (n = 31 cells); L4, 0.32 ± 1.05 (n = 35 cells); L5A, 1 ± 0.56 (n = 75 cells); L5B, 0.13 ± 0.40 (n = 50 cells); and L6, 0.29 ± 0.53 (n = 44 cells) (*Figure 3B* and *Table 4*).

The mean peak EPSP amplitude of POm input to L5A excitatory neurons therefore appears to be four times as large as input to L6 neurons, with other layers (L2, L3, L4 and L5B) having smaller amplitude EPSPs. The mean slope of EPSPs from POm appears to be roughly three times larger in L5A neurons compared to L3, L4 and L6 neurons, with L2 and L5B receiving EPSPs with even smaller slopes. Both slope and amplitude of the normalized and absolute EPSPs were significantly larger in L5A compared to any other layer (two-tailed Wilcoxon rank-sum test, $p < 10^{-7}$ for each comparison).

We also plotted EPSP amplitude and slope according to subpial distance, finding a well-defined hot-spot at a subpial depth consistent with L5A. Importantly, some neurons in other layers also received large amplitude input from POm.

## VPM input to PV neurons across layers in wS1

In order to measure VPM input onto inhibitory PV neurons, we injected ChR2-expressing virus into the VPM of PV-Cre mice (*Hippenmeyer et al., 2005*) crossed with LSL-tdTomato reporter mice (*Madisen et al., 2010*). In these mice, PV neurons are brightly fluorescent, and whole-cell recordings can readily be targeted to these neurons (*Figure 4A*). In order to compare with excitatory neurons, we also recorded from unlabeled nearby neurons. Neurons were filled with biocytin for *post hoc* anatomical analysis. We found some deep layer tdTomato-labeled neurons to be excitatory pyramidal

**Table 3.** Normalized EPSP amplitudes of POm input including: mean ± SD, *median*, and n = number of recorded cells.

| POm amplitude | EXC | PV | SST | VIP |
|---|---|---|---|---|
| L2 | 0.11 ± 0.18 *0.032* n = 38 | 0.17 ± 0.13 *0.17* n = 3 | 0.02 ± 0.02 *0.0066* n = 5 | 0.18 ± 0.37 *0.036* n = 14 |
| L3 | 0.16 ± 0.39 *0.048* n = 31 | 0.04 ± 0.04 *0.030* n = 7 | 0.07 ± 0.16 *0.016* n = 7 | 0.17 ± 0.25 *0.17* n = 20 |
| L4 | 0.13 ± 0.35 *0.011* n = 38 | 0.03 ± 0.06 *0.013* n = 8 | 0.02 ± 0.01 *0.034* n = 3 | 0.21 ± 0.35 *0.067* n = 12 |
| L5A | 1.00 ± 0.52 *1.00* n = 75 | 1.17 ± 0.96 *0.79* n = 8 | 0.15 ± 0.15 *0.053* n = 5 | 0.33 ± 0.32 *0.18* n = 10 |
| L5B | 0.11 ± 0.31 *0.026* n = 52 | 0.09 ± 0.10 *0.015* n = 10 | 0.03 ± 0.04 *0.015* n = 13 | 0.20 ± 0.18 *0.18* n = 7 |
| L6 | 0.23 ± 0.38 *0.068* n = 47 | 0.03 ± 0.05 *0.010* n = 10 | 0.01 ± 0.01 *0.010* n = 18 | *0.090* n = 2 |

**Table 4.** Normalized EPSP slopes of POm input including: mean ± SD, *median*, and n = number of recorded cells.

| POm slope | EXC | PV | SST | VIP |
|---|---|---|---|---|
| L2 | 0.08 ± 0.13 *0.014* n = 38 | 0.31 ± 0.24 *0.35* n = 3 | 0.02 ± 0.02 *0.0080* n = 4 | 0.07 ± 0.10 *0.043* n = 13 |
| L3 | 0.30 ± 1.13 *0.020* n = 31 | 0.09 ± 0.13 *0.041* n = 5 | 0.17 ± 0.41 *0.0024* n = 6 | 0.12 ± 0.12 *0.12* n = 20 |
| L4 | 0.32 ± 1.05 *0.0045* n = 35 | 0.10 ± 0.26 *0.0040* n = 8 | 0.01 ± 0.01 *0.014* n = 3 | 0.14 ± 0.20 *0.045* n = 12 |
| L5A | 1.00 ± 0.56 *1.00* n = 75 | 2.10 ± 1.96 *1.25* n = 8 | 0.24 ± 0.34 *0.045* n = 5 | 0.33 ± 0.40 *0.16* n = 10 |
| L5B | 0.13 ± 0.40 *0.014* n = 50 | 0.12 ± 0.16 *0.029* n = 10 | 0.04 ± 0.08 *0.0090* n = 13 | 0.17 ± 0.18 *0.16* n = 7 |
| L6 | 0.29 ± 0.53 *0.050* n = 44 | 0.05 ± 0.14 *0.0040* n = 10 | 0.00 ± 0.00 *0.00093* n = 14 | *0.015* n = 2 |

neurons, consistent with previous reports of PV-expressing excitatory neurons (*Hafner et al., 2019*; *Jinno and Kosaka, 2004*; *Tanahira et al., 2009*; *van Brederode et al., 1991*). These excitatory PV neurons were not included in our analyses.

The mean peak EPSP amplitude and EPSP slope in PV neurons evoked by optogenetic VPM stimulation was computed across layers (*Figure 4B*), and normalized to the L4 excitatory neurons recorded within the same slices (*Figure 4B*, *Table 1* and *Table 2*). VPM input to PV neurons in L4 was found to be about three times stronger than the input onto PV neurons in L3 for both EPSP amplitude and slope. Other layers (L2, L5A, L5B and L6) received substantially weaker input. The overall laminar pattern of VPM input is therefore similar for PV neurons and excitatory neurons. The EPSP peak amplitudes for PV neurons are also comparable to the excitatory neurons for each layer. However, the EPSP slope is larger in PV neurons compared to excitatory neurons, with the L4 PV neurons having an almost three times larger EPSP slope compared to L4 excitatory neurons. Comparing across layers, the VPM input to L4 PV neurons was significantly larger than input to PV neurons in other layers for both amplitude and slope (two-tailed Wilcoxon rank-sum test, $p < 0.05$ for each comparison).

## POm input to PV neurons across layers in wS1

We next measured POm input to inhibitory PV neurons in wS1 across layers. Similar to excitatory neurons, the largest responses in PV neurons evoked by POm stimulation were typically found in L5A, as shown in the example experiment (*Figure 5A*). We quantified the mean peak EPSP amplitude and the initial slope of the EPSPs across layers (*Figure 5B*), which we normalized to the L5A excitatory neurons recorded within the same slice (*Figure 5B*, *Table 3* and *Table 4*). POm input to PV neurons was about three times stronger in L5A compared to L2, with PV neurons in other layers (L3, L4, L5B and L6) receiving substantially less POm input. The laminar distribution of POm input to PV neurons is similar to that of excitatory neurons. Whereas the EPSP amplitude in L5A is similar in PV and excitatory neurons, the slope of the EPSP in L5A PV neurons is twice as large as that of the L5A excitatory neurons. Comparing across layers, the POm input to L5A PV neurons was significantly larger than input to PV neurons in other layers for both amplitude and slope (two-tailed Wilcoxon rank-sum test, $p < 0.05$ for each comparison).

## VPM and POm input to SST neurons across layers in wS1

In further experiments, we examined thalamic input to SST neurons. To label SST neurons, SST-Cre mice (*Taniguchi et al., 2011*) were crossed with LSL-tdTomato reporter mice. ChR2 was stereotactically injected into either VPM or POm, as before. In order to have within slice controls, we also

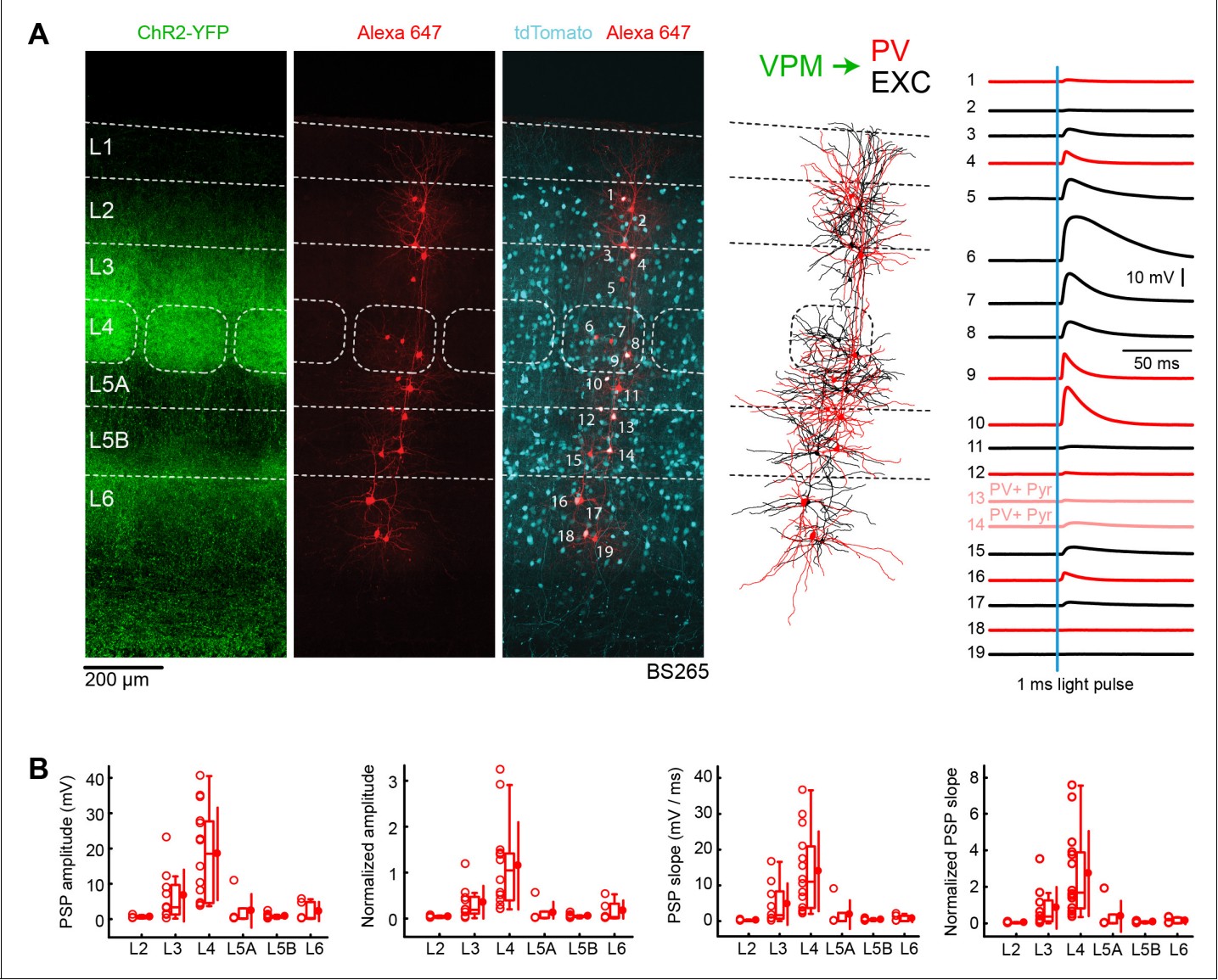

**Figure 4.** VPM input to GABAergic PV neurons across layers in wS1. (**A**) An example experiment (BS265) in which ChR2 was expressed in VPM of a PV-Cre mouse crossed with a LSL-tdTomato report mouse, and whole-cell recordings were obtained sequentially neurons across different cortical layers in wS1. In the example experiment recordings were made from 7 PV-expressing inhibitory neurons (red), 2 PV-expressing L5 pyramidal neurons (#13 and #14, pink), and 10 non-tdTomato-expressing excitatory neurons (black). Neurons were filled with biocytin and *post hoc* stained with streptavidin conjugated to Alexa 647 to reveal dendritic morphology, which was digitally reconstructed. Neurons #6 (Exc L4), #7 (Exc L4), #9 (PV L4) and #10 (PV L5A) received the largest mean EPSPs in response to optogenetic stimulation of VPM (right). (**B**) The laminar location was assigned for each recording. The peak amplitude of the evoked EPSP averaged across trials was determined for each cell, and plotted for each layer along with the median (box plot, including interquartile range and whiskers) and the mean (filled circle, along with SD error bars) (left). The EPSP amplitudes from individual slices were normalized to the mean amplitude of the EPSP measured in the L4 EXC neurons of the same slice (center left). The slope of the rising phase of the EPSP was determined and plotted for each cell across layers (center right). The normalized EPSP slope was calculated by dividing by the mean slope of the EPSPs in the L4 EXC neurons recorded in the same slice (right).

The online version of this article includes the following source data for figure 4:

**Source data 1.** Data values underlying *Figure 4*.

recorded from excitatory neurons, which we used for normalization. As shown in the example experiment, we typically only found small responses evoked by optogenetic stimulation of thalamic fibres (*Figure 6A*).

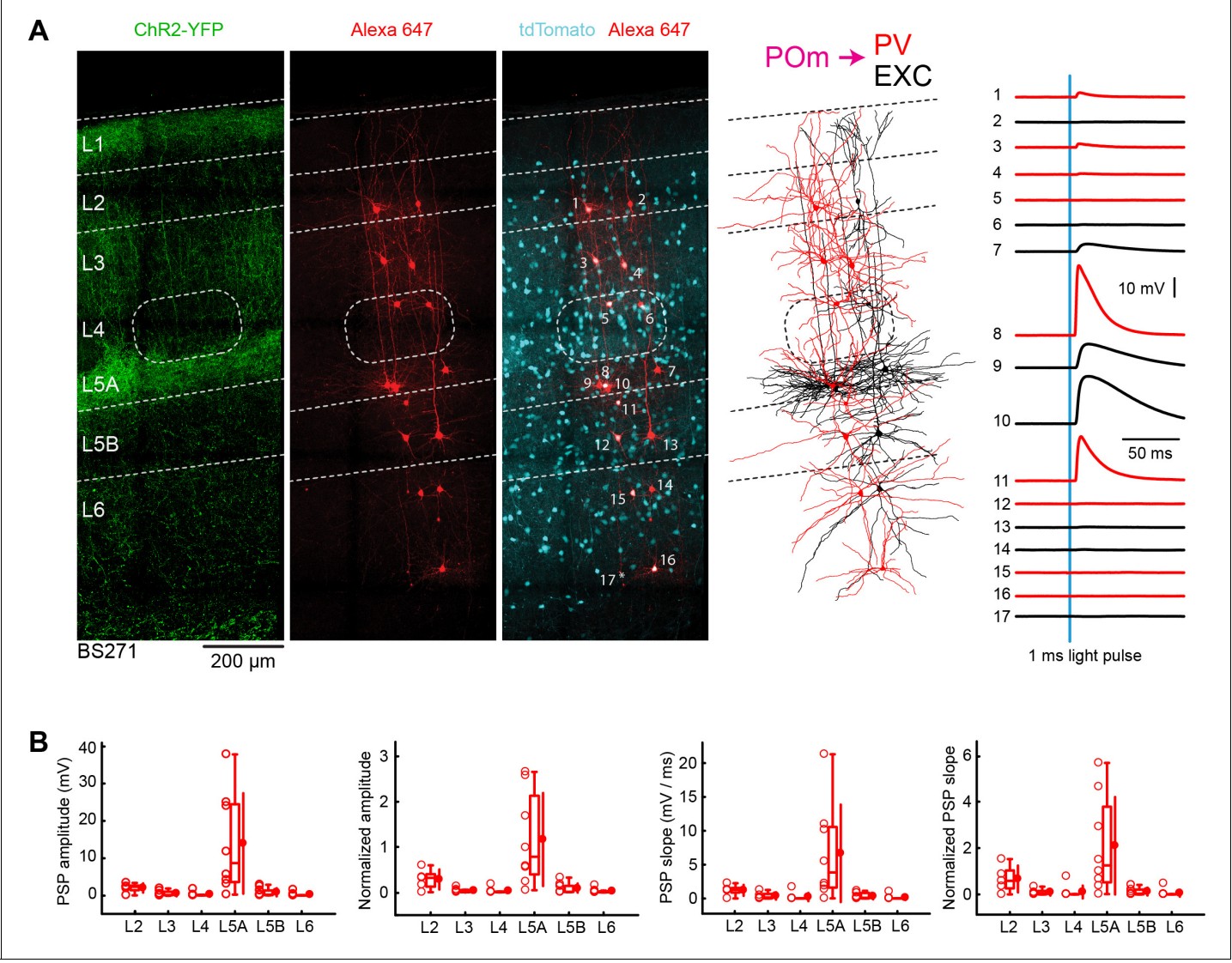

**Figure 5.** POm input to GABAergic PV neurons across layers in wS1. (**A**) An example experiment (BS271) in which ChR2 was expressed in POm and whole-cell recordings were obtained sequentially neurons across different cortical layers in wS1 of PV-Cre mice crossed with LSL-tdTomato reporter mice. In the example experiment recordings were made from 9 PV-expressing inhibitory neurons (red), and eight non-tdTomato-expressing excitatory neurons (black). Neurons were filled with biocytin and *post hoc* stained with streptavidin conjugated to Alexa 647 to reveal dendritic morphology, which was digitally reconstructed. Neurons #8 (PV L5A), #9 (Exc L5A), #10 (Exc L5A) and #11 (PV L5A) received the four largest mean EPSPs in response to optogenetic stimulation of POm (right). (**B**) The laminar location was assigned for each recording. The peak amplitude of the evoked EPSP averaged across trials was determined for each cell, and plotted for each layer along with the median (box plot, including interquartile range and whiskers) and the mean (filled circle, along with SD error bars) (left). The EPSP amplitudes from individual slices were normalized to the mean amplitude of the EPSP measured in the L5A EXC neurons of the same slice (center left). The slope of the rising phase of the EPSP was determined and plotted for each cell across layers (center right). The normalized EPSP slope was calculated by dividing by the mean slope of the EPSPs in the L5A EXC neurons recorded in the same slice (right).

The online version of this article includes the following source data for figure 5:

**Source data 1.** Data values underlying *Figure 5*.

We measured the peak EPSP amplitudes and EPSP slopes of VPM input to SST neurons across layers (*Figure 6B*), and normalized to the L4 excitatory neurons within the same slices (*Figure 6B*, *Table 1* and *Table 2*). The VPM-evoked peak EPSP amplitudes and EPSP slopes in SST neurons across all layers were small compared to the L4 excitatory neurons within the same slice. The largest

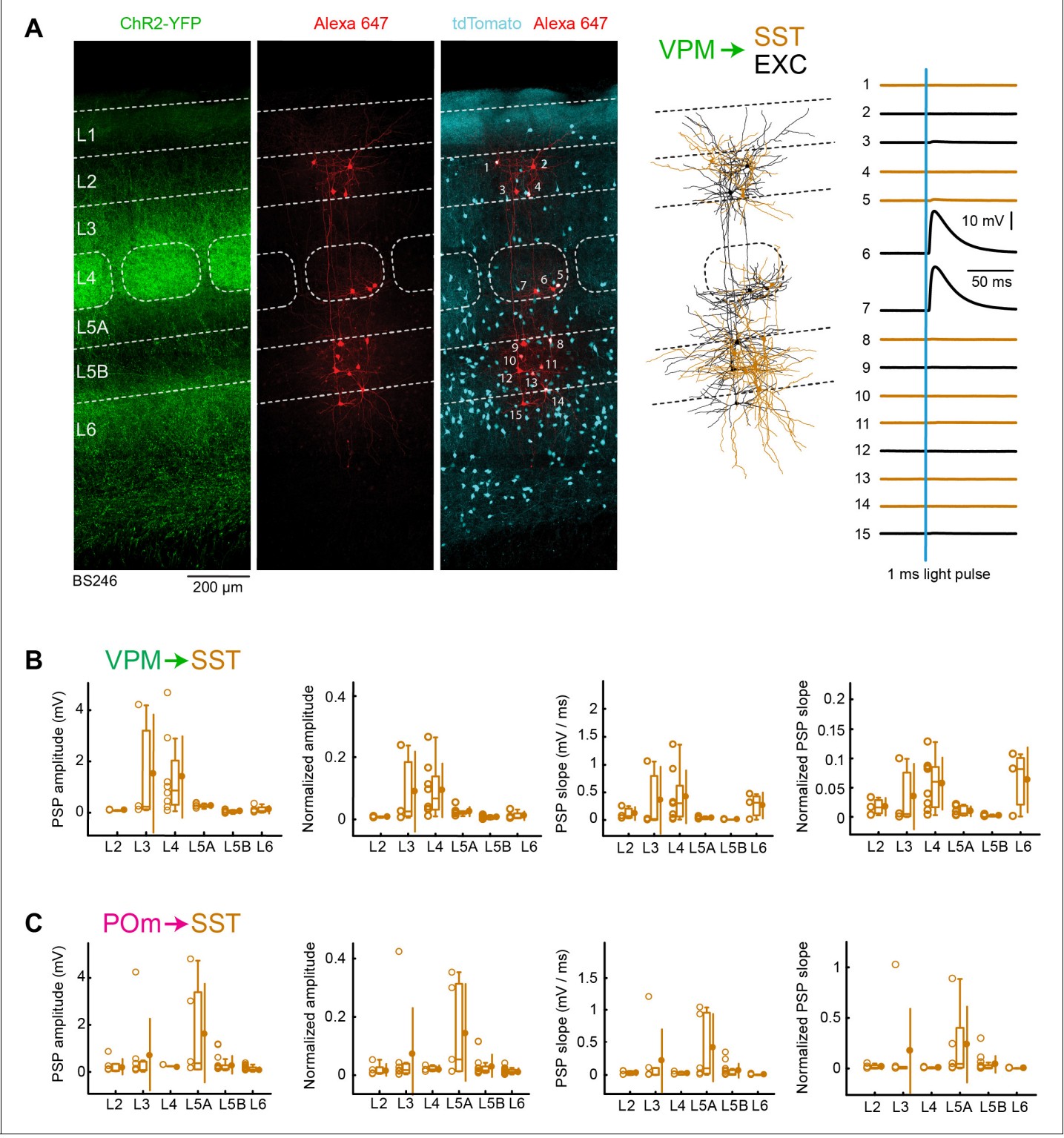

**Figure 6.** VPM and POm input to GABAergic SST neurons across layers in wS1. (**A**) An example experiment (BS246) from a SST-Cre mouse crossed with a LSL-tdTomato reporter mouse, in which ChR2 was expressed in VPM and whole-cell recordings were obtained sequentially from neurons across different cortical layers in wS1. In the example experiment recordings were made from 8 SST-expressing inhibitory neurons (red), and seven non-tdTomato-expressing excitatory neurons (black). Neurons were filled with biocytin and *post hoc* stained with streptavidin conjugated to Alexa 647 to reveal dendritic morphology, which was digitally reconstructed. Neurons #6 (EXC L4) and #7 (EXC L4) received the largest mean EPSPs in response to optogenetic stimulation of VPM (right). (**B**) The peak amplitude of the VPM-evoked EPSP averaged across trials was determined for each SST-

*Figure 6 continued on next page*

*Figure 6 continued*

expressing cell, and plotted for each layer along with the median (box plot, including interquartile range and whiskers) and the mean (filled circle, along with SD error bars) (left). The EPSP amplitudes from individual slices were normalized to the mean amplitude of the EPSP measured in the L4 EXC neurons of the same slice (center left). The slope of the rising phase of the EPSP was determined and plotted for each cell across layers (center right). The normalized EPSP slope was calculated by dividing by the mean slope of the EPSPs in the L4 EXC neurons recorded in the same slice (right). (C) As for panel B, but for POm input and normalization to L5A EXC EPSPs.

The online version of this article includes the following source data for figure 6:

**Source data 1.** Data values underlying *Figure 6*.

mean EPSP amplitudes in SST neurons were found in L3 and L4, but even these were approximately an order of magnitude smaller than the mean input to the excitatory neurons.

When analyzing POm input to SST neurons, we also found small peak EPSP amplitudes and slopes across layers (*Figure 6C*), which remained small after normalization to the L5A excitatory neurons (*Figure 6C*, *Table 3* and *Table 4*). The POm input to SST neurons was largest in L5A, but this was nonetheless more than five times smaller than the POm input to excitatory neurons in L5A.

A robust characteristic of SST neurons is that they receive facilitating excitatory input both in vitro (*Kapfer et al., 2007*; *Reyes et al., 1998*; *Silberberg and Markram, 2007*) and in vivo (*Pala and Petersen, 2015*). The small EPSP inputs to SST neurons found in our recordings could therefore result from low neurotransmitter release probability onto this cell-type, which could be enhanced through repetitive stimulation. However, we found that facilitation was absent in SST neurons under our recording conditions, presumably because optogenetic stimulation evokes large calcium spikes in the presence of TTX and 4-AP driving neurotransmitter release with high probability.

## VPM and POm input to VIP neurons across layers in wS1

Finally, we examined VPM and POm input to VIP neurons, labeled in VIP-Cre mice (*Taniguchi et al., 2011*) crossed with LSL-tdTomato reporter mice. We again recorded in parallel from excitatory neurons, in order to normalize responses within each slice. As shown in the example experiment, we typically found relatively weak inputs to VIP neurons (*Figure 7A*). However, a few VIP neurons received large (>10 mV) EPSPs in L2, L3, L4, L5A and L5B.

The mean peak EPSP amplitude and EPSP slope for VPM input to VIP neurons across layers was measured (*Figure 7B*) and normalized to the L4 excitatory neurons recorded within the same slices (*Figure 7B*, *Table 1* and *Table 2*). VIP neurons in L3 received the largest VPM input, which was roughly twice as large as the EPSP amplitude in L2, L4 and L5B.

For POm input to VIP neurons across layers, we also measured mean peak EPSP amplitude and EPSP slope (*Figure 7C*), which we normalized to the L5A excitatory neurons recorded within the same slices (*Figure 7C*, *Table 3* and *Table 4*). POm input to VIP neurons appeared to be distributed more-or-less evenly across layers, with a somewhat larger mean EPSP in L5A.

## Discussion

Our measurements begin to quantify layer- and cell-type-specific input from VPM and POm to wS1 (*Figure 8* and *Tables 1–4*). Our data suggest that VPM and POm input to excitatory and PV neurons is much stronger than the thalamic input to SST neurons, with VIP neurons receiving an intermediate level. Input to excitatory, PV and SST neurons largely appears to follow the pattern of axonal innervation density, with VPM most strongly innervating L4 (and L3), and POm most strongly innervating L5A. VIP neurons appear to receive thalamic input with less layer-dependence.

### Layer-dependent thalamic input to excitatory neurons in wS1

VPM and POm have complementary wS1 innervation patterns (*Figure 1A*) (*Wimmer et al., 2010*), which are clearly reflected in the layer-specific average EPSP input to excitatory neurons (*Figures 2*, *3* and *8*). Consistent with the pathway-specific anatomical axonal projections, excitatory neurons in L4 received the largest VPM input, with neurons in L3 on average also receiving prominent VPM input. L5A neurons on average received the smallest VPM input. In contrast, L5A neurons on average received much stronger POm input compared to excitatory neurons in any other layer. Our results

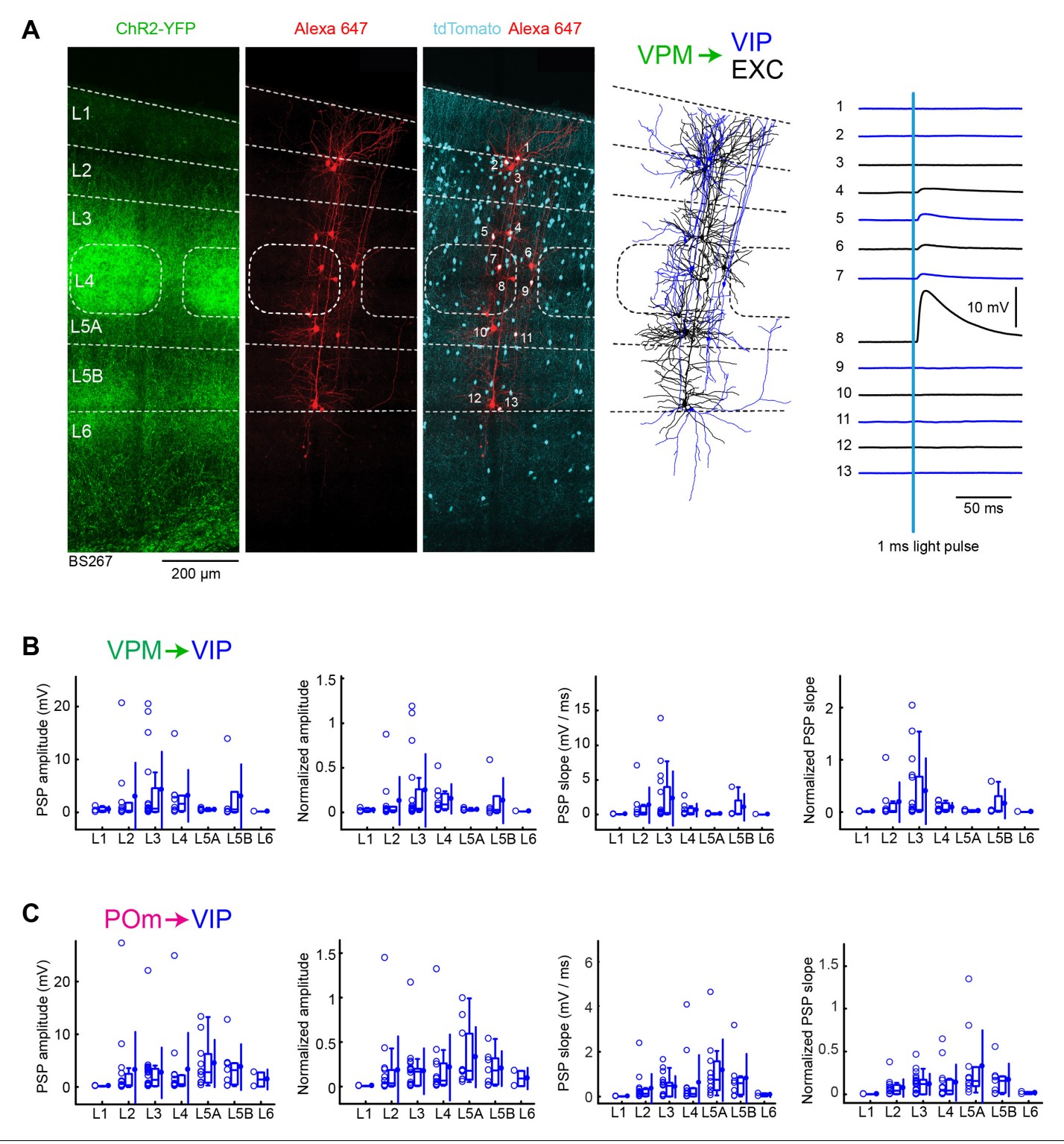

**Figure 7.** VPM and POm input to GABAergic VIP neurons across layers in wS1. (**A**) An example experiment (BS267) in which ChR2 was expressed in VPM of a VIP-Cre mouse crossed with a LSL-tdTomato reporter mouse, and whole-cell recordings were obtained sequentially from neurons across different cortical layers in wS1. In the example experiment recordings were made from 7 VIP expressing inhibitory neurons (red), and six non-tdTomato-expressing excitatory neurons (black). Neurons were filled with biocytin and *post hoc* stained with streptavidin conjugated to Alexa 647 to reveal dendritic morphology, which was digitally reconstructed. Neuron #8 (Exc L4) received the largest mean EPSP in response to optogenetic stimulation of VPM (right). (**B**) The peak amplitude of the VPM-evoked EPSP averaged across trials was determined for each VIP-expressing cell, and plotted for each

*Figure 7 continued on next page*

*Figure 7 continued*

layer along with the median (box plot, including interquartile range and whiskers) and the mean (filled circle, along with SD error bars) (left). The EPSP amplitudes from individual slices were normalized to the mean amplitude of the EPSP measured in the L4 EXC neurons of the same slice (center left). The slope of the rising phase of the EPSP was determined and plotted for each cell across layers (center right). The normalized EPSP slope was calculated by dividing by the mean slope of the EPSPs in the L4 EXC neurons recorded in the same slice (right). (C) As for panel B, but for POm input and normalization to L5A EXC neurons.

The online version of this article includes the following source data for figure 7:

**Source data 1.** Data values underlying *Figure 7*.

are in good agreement with previous measurements (*Agmon and Connors, 1992*; *Audette et al., 2018*; *Bureau et al., 2006*; *Cruikshank et al., 2010*; *Petreanu et al., 2009*).

In addition to highlighting largely segregated VPM and POm input to different layers, our data also suggest that there could be convergence of these two pathways in L6, which on average received a normalized VPM input strength of 0.39 (*Table 1*) and a normalized POm input strength of 0.23 (*Table 3*). Future experiments should probe whether this is onto the same or different L6 cells, which would give rise to different functional implications and hypotheses.

Although there are clear layer-specific differences in synaptic inputs on average, within each layer there were typically some excitatory neurons that received much larger than average thalamic input. Thalamic input therefore directly influences neurons in all layers of wS1. In future studies, it will be of great interest to investigate the variability of input to excitatory neurons within the same layer, which could relate to experience-dependent synaptic plasticity (*Audette et al., 2019*; *Oberlaender et al., 2012b*) or different subtypes of excitatory neurons, for example classified according to their axonal projections (*Chen et al., 2013*; *Harris and Shepherd, 2015*; *Rojas-Piloni et al., 2017*; *Yamashita et al., 2013*).

Thalamic input from VPM is thought to dominate the earliest wS1 sensory responses evoked by whisker deflection (*Bruno and Sakmann, 2006*; *Cohen-Kashi Malina et al., 2016*; *Gutnisky et al., 2017*; *Oberlaender et al., 2012a*; *Pinto et al., 2000*). The layer specific VPM input measured here corresponds well to measurements of whisker-deflection evoked PSPs from in vivo whole-cell membrane potential recordings. Neurons in L4 of wS1 on average depolarize in response to whisker deflection with shorter latencies and larger amplitudes than neurons in other layers (*Brecht et al., 2003*; *Brecht and Sakmann, 2002*; *Manns et al., 2004*; *Moore and Nelson, 1998*; *Zhu and Connors, 1999*). During active touch, neurons in deeper parts of L2/3 on average depolarize more rapidly and strongly than neurons in superficial L2/3 (*Crochet et al., 2011*). Nonetheless, VPM input to subsets of neurons in L5 can be strong and fast (*Figure 2*) (*Constantinople and Bruno, 2013*), and in the future it will be important to investigate which subtypes of L5 pyramidal neurons receive strong VPM input.

Sufficiently large thalamic excitatory synaptic input can drive membrane potential across the action potential threshold to evoke spiking activity. Consequently layer-specific thalamic input is also reflected in the timing of whisker-deflection-evoked action potentials measured with extracellular recordings. A brief whisker deflection evokes action potential firing with shortest latency in L4 and L5B (*Armstrong-James et al., 1992*; *Constantinople and Bruno, 2013*; *de Kock et al., 2007*). During texture sampling, whiskers exhibit rapid slips as they overcome surface friction, and these slips evoke rapid sensory responses in wS1, with action potentials in L4 being evoked at the shortest latencies (*Isett et al., 2018*).

## Strong thalamic input to PV neurons

PV neurons received strong thalamic input with an overall similar layer-dependence to that of the excitatory neurons. PV neurons in L4 (and L3) received the largest VPM input, and PV neurons in L5A received the largest POm input, in good agreement with previous studies (*Audette et al., 2018*; *Bagnall et al., 2011*; *Cruikshank et al., 2010*). The peak EPSP amplitudes evoked in PV neurons was similar to that measured for the excitatory neurons in the same layer. The EPSP slope however was typically much faster in PV neurons compared to the excitatory neurons, perhaps because of large, fast synaptic conductances and short membrane time-constants typical of cortical PV-expressing fast-spiking GABAergic neurons (*Hu et al., 2014*). The large and rapid thalamic input to PV neurons, likely enables rapid feedforward inhibition in the local wS1 microcircuit (*Cruikshank et al.,*

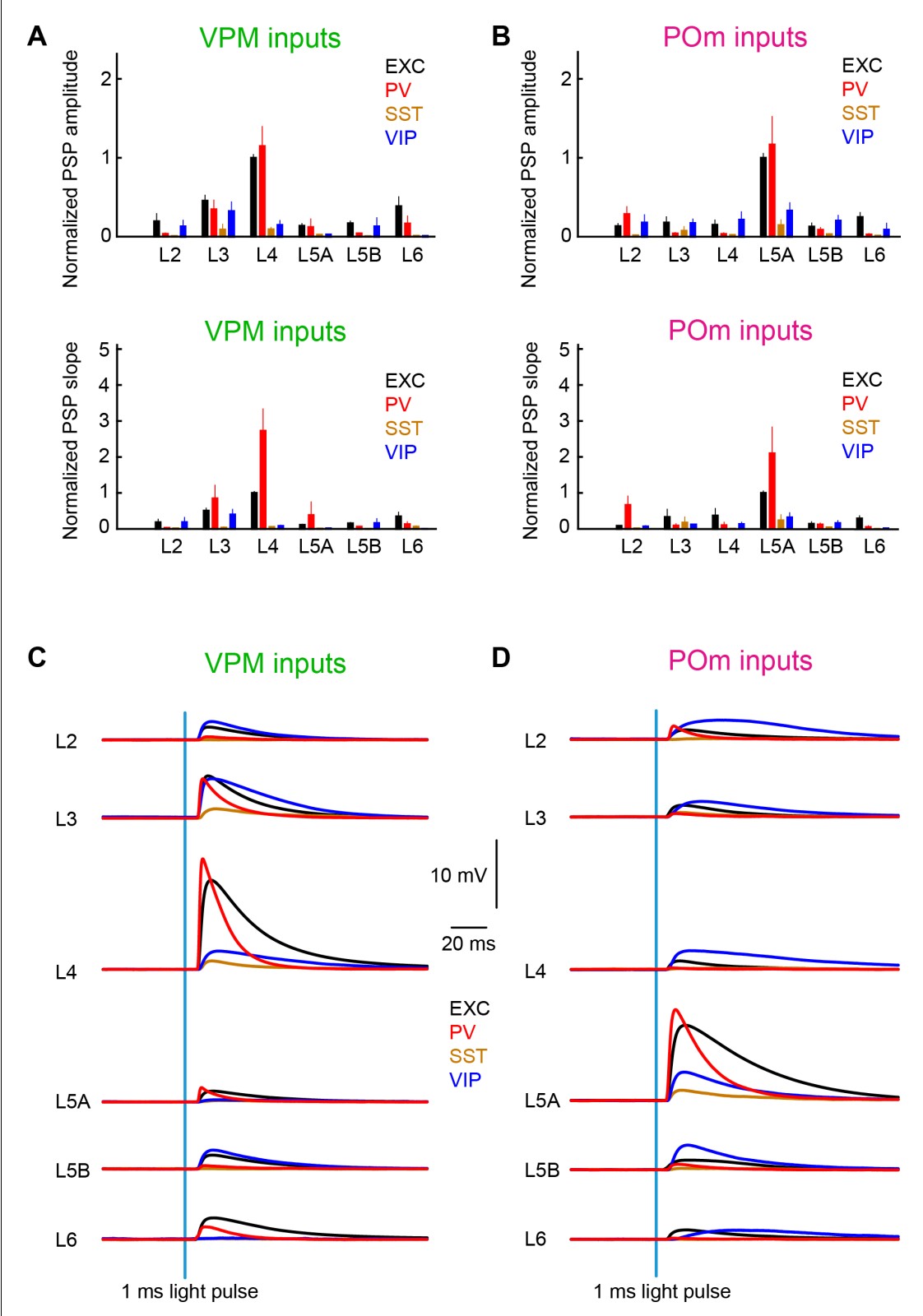

**Figure 8.** Summary of layer-specific inputs from VPM and POm to distinct cell-types in wS1. (**A**) The mean ± SEM of EPSP amplitudes across layers (normalized to L4 EXC) for excitatory neurons (black), PV neurons (red), SST neurons (brown) and VIP neurons (blue) (above). The equivalent plot for the EPSP slope is shown below. (**B**) As for panel A, but for POm inputs normalized to L5A EXC. (**C**) The mean time-course of EPSPs evoked by the 1 ms

*Figure 8 continued on next page*

*Figure 8 continued*

blue light pulse in excitatory neurons (black), PV neurons (red), SST neurons (brown) and VIP neurons (blue) separately averaged across layers. (**D**) As for panel C, but for POm inputs.

*2010*; *Cruikshank et al., 2007*; *Gabernet et al., 2005*), which partially may serve to suppress circuit activity during exploratory whisking (*Yu et al., 2016*).

In vivo measurements of L2/3 neuronal activity evoked by a brief whisker stimulus suggest that PV neurons, as compared to other types of nearby L2/3 neurons: i) respond with the shortest latency, ii) have the largest amplitude PSPs, and iii) have the largest increase in action potential firing (*Gentet et al., 2012*; *Sachidhanandam et al., 2016*). The large and fast sensory responses in PV neurons could be driven by large amplitude and fast VPM thalamic input to PV neurons, consistent with our brain slice measurements (*Figure 4*).

The rapid recruitment of GABAergic inhibition in response to whisker stimulation is likely responsible for the overall sparse action potential firing of excitatory neurons in wS1 (*Barth and Poulet, 2012*; *Jouhanneau et al., 2018*; *O'Connor et al., 2010*; *Petersen, 2019*). Most excitatory neurons in wS1 will receive near simultaneous excitatory glutamatergic input and inhibitory GABAergic input. A brief whisker stimulus drives most individual neurons towards a reversal potential, which is typically hyperpolarized relative to action potential threshold (*Crochet et al., 2011*; *Sachidhanandam et al., 2013*). Only a sparse population of cortical neurons receiving sufficiently large and fast excitation typically reliably fire action potentials and reliably encode the sensory stimulus (*Crochet et al., 2011*; *Sachidhanandam et al., 2013*). In future measurements, it will be of interest to compare the thalamic input, as well as local wS1 microcircuit connectivity, of the neurons responding strongly to whisker stimulation relative to their weakly responding neighbors.

Thalamic action potential firing rates in both VPM and POm increase during whisking compared to during periods of quiet wakefulness without whisker movements, with the most prominent increase being in VPM neurons (*Moore et al., 2015*; *Poulet et al., 2012*; *Urbain et al., 2015*). The thalamic action potential firing during whisking appears to contribute importantly to driving an active desynchronized cortical state (*Poulet et al., 2012*). The increased firing rates of thalamic neurons during whisking will strongly excite both excitatory and PV neurons, likely driving the cortical circuits to a cell-specific distribution of reversal potentials with a reduced membrane potential variance over time compared to during quiet wakefulness, consistent with experimental measurements (*Gentet et al., 2010*; *Pala and Petersen, 2018*; *Poulet and Petersen, 2008*; *Yu et al., 2019*; *Yu et al., 2016*).

Our results are consistent with a large body of literature suggesting that PV neurons are likely to play a profound role for the fast balance of excitation and inhibition in local cortical microcircuits (*Freund and Katona, 2007*; *Hu et al., 2014*; *Isaacson and Scanziani, 2011*; *Tremblay et al., 2016*).

## Diverse thalamic input onto VIP neurons

Most VIP neurons received relatively little thalamic input, but in each layer we found some strongly innervated VIP neurons. In contrast to excitatory and PV neurons, the VIP neurons on average appear to receive thalamic input with less dependence upon the location of the cell body. Many VIP neurons have extensive vertically-oriented dendritic arborizations spanning several cortical layers and their relatively high input resistance might allow their somata to integrate even the thalamic input arriving in distal dendrites. In future studies, it will be important to correlate the subtype of VIP neuron (*He et al., 2016*; *Prönneke et al., 2019*; *Prönneke et al., 2015*) with the amount of thalamic input that it receives.

Functionally, VIP neurons in L2/3 depolarize and increase firing rate during whisking (*Lee et al., 2013*), which might in part result from the increased firing rate of VPM and POm neurons during whisking, as well as other inputs, including cholinergic input which might depolarize VIP neurons via ionotropic receptor activation (*Fu et al., 2014*). VIP neurons also respond well to whisker deflection, although at longer latencies than PV neurons (*Sachidhanandam et al., 2016*). VIP neurons strongly innervate SST neurons (*Karnani et al., 2016*; *Lee et al., 2013*; *Pfeffer et al., 2013*; *Pi et al., 2013*; *Walker et al., 2016*), and it is likely that the increased firing rates of VIP neurons during whisking and sensory processing strongly impact the activity of SST neurons, as discussed below.

## Weak thalamic input onto SST neurons

Consistent with most previous reports, we found that thalamic input from both VPM and POm was weak onto SST neurons (*Audette et al., 2018*; *Cruikshank et al., 2010*). Weak VPM input to SST neurons in wS1 might account for their relative lack of strong sensory-evoked responses, at least in L2/3 (*Gentet et al., 2012*; *Sachidhanandam et al., 2016*). Sensory responses are likely further suppressed in SST neurons because they receive a prominent inhibitory input from VIP neurons (*Karnani et al., 2016*; *Lee et al., 2013*; *Pfeffer et al., 2013*; *Pi et al., 2013*; *Walker et al., 2016*), which respond strongly to whisker stimulation (*Sachidhanandam et al., 2016*).

The spontaneous membrane potential fluctuations of L2/3 SST neurons are also relatively decoupled from the local slow (1–10 Hz) synchronous oscillations of other nearby wS1 neurons (*Gentet et al., 2012*; *Pala and Petersen, 2018*). These slow oscillations appear in part to be driven by thalamic activity (*Poulet et al., 2012*), and the relative paucity of thalamic input to SST neurons is likely to contribute to their relative lack of synchronous slow oscillations. During the active cortical state during whisking, L2/3 SST neurons hyperpolarize (*Gentet et al., 2012*; *Pala and Petersen, 2018*). This is likely in part because they do not receive strong thalamic input (which increases during whisking) and in part because they receive strong inhibition from VIP neurons (which increase firing during whisking).

SST neurons prominently innervate distal dendrites of excitatory pyramidal neurons (*Zhou et al., 2020*), and the disinhibitory circuit of VIP neurons suppressing SST neurons, might play an important role in releasing distal dendrites from inhibition during sensory processing and sensorimotor integration (*Gentet et al., 2012*). Disinhibition is also likely to play an important role for associative synaptic plasticity (*Williams and Holtmaat, 2019*).

## Future perspectives

There are many important limitations in the current study, and much further work remains to be done before a quantitative wiring diagram of thalamic input to wS1 is established. The current study involves recordings from a small number of neurons in relationship to the number of neurons in any given cortical column, and involves comparisons across animals. In the future it will be important to establish more complete wiring diagrams for single animals, which might be helped by the development of electron microscopy connectomics (*Helmstaedter et al., 2013*; *Kasthuri et al., 2015*; *Lee et al., 2016*) and new optical methods for stimulating large numbers of individual neurons through multiphoton optogenetics (*Mardinly et al., 2018*; *Packer et al., 2015*; *Prakash et al., 2012*; *Shemesh et al., 2017*).

Another important limitation in terms of the physiological interpretation of our results stems from our choice to block voltage-gated sodium channels with TTX application and voltage-gated potassium channels with 4-AP in order to drive synaptic release evoked by ChR2 stimulation rather than to rely upon action potential evoked release. Under our recording conditions, we found this was essential to block polysynaptic activity, which would have impacted our ability to quantify monosynaptic thalamic input. In the future, it would be of great interest to study thalamic input evoked by optical minimal stimulation (*Morgenstern et al., 2016*), which might allow quantification of unitary thalamic inputs onto the various types of neurons in wS1.

The brain slice preparation inevitably involves truncation of dendritic and axonal arborizations. Our study therefore likely underestimates the actual amount of thalamic input, and the in vivo connectivity strength of thalamic input is likely to be higher than reported here. In the future, it would therefore be important to make in vivo measurements across layers studying VPM and POm input onto different cell-types (*Bruno and Sakmann, 2006*; *Constantinople and Bruno, 2013*; *Zhang and Bruno, 2019*). In addition, because of the complex three-dimensional structure, in brain slices it is difficult to identify the precise location of recorded neurons within the horizontal extent of individual barrel columns and their relationship to septa, which are very small in mice compared to rats. Here, in this study, we thus did not attempt to assign horizontal locations of neurons, but, in the future, in vivo measurements could help more precisely localize recorded neurons within the intact barrel map.

All the whole-cell recordings in the current study were targeted to the cell body. Given that synaptic potentials are strongly attenuated across dendritic arborizations (*Nevian et al., 2007*), it is likely that our measurements largely report synaptic input onto proximal dendrites. Important

computations appear to take place in dendrites (*Lavzin et al., 2012*; *Ranganathan et al., 2018*; *Smith et al., 2013*; *Takahashi et al., 2016*), and detailed synaptic input maps across dendritic arborisations (*Petreanu et al., 2009*) will be essential to more fully understand the impact of thalamic input onto cortical neurons. The relative importance of distal dendritic synaptic input might also differ comparing in vivo and in vitro experimental conditions. The more depolarized state of dendrites in vivo driven by ongoing synaptic input could contribute importantly to amplifying thalamic input, perhaps providing a mechanism for increasing the efficacy of the, on average, relatively weak VPM input to L5B excitatory pyramidal neurons found in vitro in this study (*Figure 2*) compared to the prominent role of VPM input to L5B neurons found in vivo (*Constantinople and Bruno, 2013*).

Single-cell gene expression analysis has revealed a large number of distinct classes of neurons (*Tasic et al., 2018*; *Tasic et al., 2016*; *Zeisel et al., 2018*; *Zeisel et al., 2015*). In the future it will be of enormous importance to specifically investigate the thalamic input onto more precisely defined groups of neurons. Furthermore, in this study we have entirely omitted several classes of inhibitory neurons, including the large group of 5HT3A-receptor expressing non-VIP neurons (of which neurogliaform cells are a subset), and we have not studied the inhibitory neurons in L1 (*Schuman et al., 2019*). The development of intersectional genetic strategies for more precisely targeting inhibitory neurons is of enormous importance (*He et al., 2016*), and will enable functional measurements. Along with better sub-typing of inhibitory neurons, it is also important to further subdivide the excitatory neurons, which might be carried out according to long-range projection targets through retrograde labeling. Indeed, previous work has already found that local intracortical synaptic connectivity differs according to the long-range projections of excitatory neurons (*Anderson et al., 2010*; *Brown and Hestrin, 2009*; *Kim et al., 2018*), and it is thus possible that there might also by projection-specific differences in thalamic inputs. There is therefore a very large amount of work to be done before a complete understanding of thalamic innervation of wS1 can be claimed. An essential further step is to integrate quantitative thalamocortical connectivity data into computational models of wS1 microcircuits in order to better understand cortical function. Understanding cell-type-specific neocortical function is important because different genetically-defined classes of neurons express different gene-products providing potential targets for cell-type-specific drugs and therapies for diverse brain disorders.

# Materials and methods

**Key resources table**

| Reagent type (species) or resource | Designation | Source or reference | Identifiers | Additional information |
|---|---|---|---|---|
| Strain, strain background (*Mus musculus*) | *Short: Wild type* C57BL/6J | Jackson Laboratory | JAX:000664 | |
| Genetic reagent (*Mus musculus*) | *Short: GPR26-Cre* Tg(Gpr26-cre) KO250Gsat/Mmucd | MMRRC | RRID: MMRRC_033032-UCD | (*Gerfen et al., 2013*) |
| Genetic reagent (*Mus musculus*) | *Short: PV-Cre* B6;129P2-Pvalb$^{tm1(cre)Arbr}$/J | Jackson Laboratory | JAX:008069 | (*Hippenmeyer et al., 2005*) |
| Genetic reagent (*Mus musculus*) | *Short: Sst-Cre* Sst$^{tm2.1(cre)Zjh}$/J | Jackson Laboratory | JAX:013044 | (*Taniguchi et al., 2011*) |
| Genetic reagent (*Mus musculus*) | *Short: VIP-Cre* Vip$^{tm1(cre)Zjh}$/J | Jackson Laboratory | JAX:010908 | (*Taniguchi et al., 2011*) |
| Genetic reagent (*Mus musculus*) | *Short: Gad2-Cre* Gad2$^{tm2(cre)Zjh}$/J | Jackson Laboratory | JAX:010802 | (*Taniguchi et al., 2011*) |
| Genetic reagent (*Mus musculus*) | *Short: LSL-tdTomato* B6.Cg-Gt(ROSA) 26Sor$^{tm9(CAG-tdTomato)Hze}$/J | Jackson Laboratory | JAX:007909 | (*Madisen et al., 2010*) |
| Recombinant DNA reagent | AAV-FLEX-tdTomato | Addgene | Addgene #28306 | |

*Continued on next page*

*Continued*

| Reagent type (species) or resource | Designation | Source or reference | Identifiers | Additional information |
|---|---|---|---|---|
| Recombinant DNA reagent | AAV-hSyn-ChR2-YFP | U. Penn | Addgene #26973 | |
| Recombinant DNA reagent | AAV-DIO-ChR2-YFP | U. Penn | Addgene #20298 | |
| Recombinant DNA reagent | AAV-DFO-ChR2-YFP | *This paper* | Addgene #136916 | See Materials and methods Available from Addgene |
| Software, algorithm | Matlab analysis code and data | *This paper* | Zenodo DOI: 10.5281/zenodo.3560697 | See Materials and methods Available from https://zenodo.org/ |

## Authorization for animal experiments

All experiments were performed in accordance with the Swiss Federal Veterinary Office, under authorization 1889 issued by the 'Service de la consommation et des affaires vétérinaires' of the Canton de Vaud, Switzerland.

## Preparation of Cre-OFF expression vector

A Cre-dependent AAV vector for Cre-OFF expression (double-floxed open reading frame, DFO) was generated by cloning the ChR2-eYFP transgene in the forward orientation immediately downstream to the human synapsin-1 (hSyn1) promoter (Addgene plasmid #136916). In this configuration, Cre activity reverses the orientation of the transgene such that it is no longer transcribed. Viral vectors were packaged in HEK cells using the hybrid AAV1/2 capsid system (*Grimm et al., 2003*). Briefly, HEK293 cells were transfected with plasmids encoding AAV rep, cap of AAV1 and AAV2 and the vector plasmid described above using the polyethylenimine (PEI) method. Cells and medium were harvested 72 hr after transfection, pelleted by centrifugation (300 g), resuspended and lysed. Crude lysate was treated with 250 U benzonase (Sigma) per 1 ml of lysate at 37°C for 1.5 hr and centrifuged at 3,000 g for 15 min. AAV particles in the supernatant were purified using heparin-agarose columns, eluted with soluble heparin, washed with phosphate buffered saline (PBS) and concentrated by Amicon columns to yield viral titers of $\sim 10^{11}$ genome copies per milliliter (gc/ml).

## ChR2 virus injection in the thalamus

Virus injections were targeted to VPM (1.6 mm posterior to Bregma; 2.0 mm lateral to the midline; vertical depth 3.25 mm from pial surface) or POm (2.0 mm posterior to Bregma; 1.25 mm lateral to the midline; vertical depth 2.8 mm from pial surface) under deep isofluorane anesthesia. Approximately 50 nl of different types of adeno-associated viruses encoding ChR2-eYFP were injected with a thin glass pipette (diameter $\sim$20 µm). AAV2/5.EF1$\alpha$.DIO.hChR2(H134R).eYFP (Penn Vector Core) and AAV2/1.hSyn.DFO.ChR2.eYFP (see above) were used to express ChR2 in the POm or VPM nucleus of the thalamus of GPR26-Cre mice. AAV2/5.hSyn.hChR2(H134R).eYFP (Addgene plasmid #26973) was used to express ChR2 in the thalamus of PV-Cre x LSL-tdTomato, SST-Cre x LSL-tdTomato, VIP-Cre x LSL-tdTomato mice and GAD2-Cre x LSL-tdTomato. Injections were made when the mice are 3–4 weeks old. After injection mice were returned to their home cages for 4–8 weeks to allow time for expression.

## Brain slice preparation

The brains of GPR26-Cre, PV-Cre x LSL-tdTomato, SST-Cre x LSL-tdTomato, VIP-Cre x LSL-tdTomato and GAD2-Cre x LSL-tdTomato mice of either sex were removed at postnatal days 42–92 (4–8 weeks after viral injections), and 300-µm-thick parasagittal (35° away from vertical) brain slices were cut on a vibrating slicer (Leica VT1200S) in an ice-cold modified artificial cerebrospinal fluid (ACSF) containing (in mM) 87 NaCl, 25 NaHCO$_3$, 25 D-glucose, 2.5 KCl, 1.25 NaH$_2$PO$_4$, 0.5 CaCl$_2$, 7 MgCl$_2$, 75 Sucrose, aerated with 95% O$_2$ + 5% CO$_2$. After being sliced, the tissue was transferred to a chamber with the same solution at room temperature for 25 min. Then the tissue was transferred to a

chamber with standard ACSF, containing (in mM) 125 NaCl, 25 $NaHCO_3$, 25 D-glucose, 2.5 KCl, 1.25 $NaH_2PO_4$, 2 $CaCl_2$, 1 $MgCl_2$, aerated with 95% $O_2$/5% $CO_2$ at room temperature. Slices were maintained at room temperature until the recording session started (within 3 hr of slicing).

## In vitro whole-cell recordings

The brain slices containing ChR2-YFP expressing axons were identified with a 4x objective lens (Olympus UPlanFI 4x, 0.13 NA) using very brief illumination with 470 nm light to excite YFP fluorescence. tdTomato-expressing GABAergic neurons were visualized using illumination with 580 nm light to excite tdTomato fluorescence. Excitation light was focused into the slice tissue with a 40 × 0.8 NA water-immersion objective (Olympus). Creation of a gradient contrast image of unlabeled cells was achieved by transmitted light through a Dodt contrast element (Luigs and Neumann). Brain slices were continually superfused with ACSF with 50 µM picrotoxin (PTX), 1 µM tetrodotoxin (TTX), 100 µM 4-aminopyridine (4-AP) at 34°C and aerated with 95% $O_2$-5% $CO_2$.

Neurons were recorded in the whole-cell configuration with a Multi-clamp 700B amplifier (Molecular Devices). Borosilicate patch pipettes with resistance of 5–7 MΩ were used. The pipette intracellular solution contained (in mM) 135 K-gluconate, 4 KCl, 4 Mg-ATP, 10 $Na_2$-phosphocreatine, 0.3 Na-GTP, and 10 HEPES (pH 7.3, 280 mOsmol/l). Biocytin (3 mg/ml) was added to intracellular solution. Electrophysiological data were low-pass Bessel filtered at 10 kHz and digitized at 20 kHz with an ITC-18 acquisition board (Instrutech). Data acquisition routines were custom-made procedures written in IgorPro software (Wavemetrics). Membrane potential measurements were not corrected for the liquid junction potential.

For stimulation of ChR2-expressing axons, we used a 470 nm collimated blue LED system (Thorlabs) coupled to a 1 mm optic fiber (Thorlabs; 0.48NA), which was directed at the cortical region targeted for optogenetic stimulation by bringing the end of the fiber immediately next to the brain slice. Optic fiber blue light stimulation (1 ms pulses) had a peak light power which varied across experiments between ~1 mW - ~ 30 mW (maintained constant for the data analyzed in each slice).

## Anatomy

After completion of the electrophysiological recordings, slices were fixed for at least 24 hr in 4% paraformaldehyde and then transferred into phosphate buffer saline (PBS). Slices were then washed in PBS three times over a period of 1 hr. After washing, slices were then incubated in blocking solution containing 5% normal goat serum (NGS) and 0.3% Triton X-100 for 1 hr. Then slices were transferred into the staining solution containing 0.3% Triton X-100 and 1:2000 of Streptavidin conjugated to Alexa 647 (Life Technologies). Slices were incubated for 3 hr and then washed with PBS at room temperature. DAPI was used as a counterstain. Slices were then mounted and imaged under a confocal microscope (Leica SP8 FLIM) through a 25x/0.95NA water objective (HC Fluotar) or 63x/1.30NA glycerol objective (HC PL APO). All the recovered neurons could be identified and matched to the recording. Vertical cell depth from the pial surface was measured on the fixed slice. In the cases where the cell could not be recovered, the manipulator reading was taken as the depth.

## Data analysis

Electrophysiological data were analyzed using custom-made routines written in Matlab (MathWorks). The data and Matlab analysis code for generating the figures are freely available at the CERN database Zenodo https://zenodo.org/communities/petersen-lab-data with DOI: 10.5281/zenodo.3560697. Mean traces were calculated by averaging over 20 single trials. Membrane potential traces were aligned to the onset of the 1 ms ChR2 stimulus of the thalamocortical axons. Mean PSP amplitudes were calculated by taking the average peak and subtracting the baseline pre-stimulus membrane potential. PSP baselines were defined as the mean $V_m$ 100 ms before stimulus onset. The slope of the PSP was calculated from a linear fit to the 20–50% rise-time period.

Population data are represented as mean ± SD in scatter plots. In box plots, the median and interquartile range are shown with whiskers extending from the smallest data point comprised within 1.5x of the interquartile range of the first quartile to the largest data point comprised within 1.5x of the interquartile range of the third quartile.

The Wilcoxon signed-rank test was used to compare two groups of paired data in *Figure 1B* and *Figure 1C*.

## Acknowledgements

We thank Matthieu Auffret, Sami El-Boustani and Varun Sreenivasan for discussions. This work was supported by the Swiss National Science Foundation (310030B_166595 to CCHP), the European Research Council (ERC-2011-ADG 293660 to CCHP and ERC-2018-CoG 819496 to OY) and the Deutsche Forschungsgemeinschaft via the CRC 889 'Cellular mechanisms of sensory processing' (TP C07 to JFS). We thank Nicolas Zdun, Nils Lammers and Johanna Rieke for Neurolucida reconstructions.

## Additional information

### Funding

| Funder | Grant reference number | Author |
| --- | --- | --- |
| Swiss National Science Foundation | 310030B_166595 | Carl CH Petersen |
| European Research Council | ERC-2011-ADG 293660 | Carl CH Petersen |
| Deutsche Forschungsgemeinschaft | CRC 889 'Cellular mechanisms of sensory processing' TP C07 | Jochen F Staiger |
| European Research Council | ERC-2018-CoG 819496 | Ofer Yizhar |

The funders had no role in study design, data collection and interpretation, or the decision to submit the work for publication.

### Author contributions

B Semihcan Sermet, Conceptualization, Data curation, Software, Formal analysis, Investigation, Visualization, Methodology; Pavel Truschow, Michael Feyerabend, Conceptualization, Software, Formal analysis, Investigation, Visualization, Methodology; Johannes M Mayrhofer, Software, Formal analysis, Visualization, Methodology; Tess B Oram, Ofer Yizhar, Resources, Methodology; Jochen F Staiger, Conceptualization, Supervision, Funding acquisition, Visualization; Carl CH Petersen, Conceptualization, Supervision, Funding acquisition

### Author ORCIDs

Ofer Yizhar http://orcid.org/0000-0003-4228-1448
Carl CH Petersen https://orcid.org/0000-0003-3344-4495

### Ethics

Animal experimentation: All experiments were performed in accordance with the Swiss Federal Veterinary Office, under authorization 1889 issued by the 'Service de la consommation et des affaires vétérinaires' of the Canton de Vaud.

### Decision letter and Author response

Decision letter https://doi.org/10.7554/eLife.52665.sa1
Author response https://doi.org/10.7554/eLife.52665.sa2

## Additional files

### Supplementary files

• Transparent reporting form

### Data availability

The data and Matlab analysis code for generating the figures are freely available at the CERN database Zenodo https://zenodo.org/communities/petersen-lab-data with DOI: https://doi.org/10.5281/zenodo.3560697.

The following dataset was generated:

| Author(s) | Year | Dataset title | Dataset URL | Database and Identifier |
|---|---|---|---|---|
| B Semihcan Sermet, Pavel Truschow, Michael Feyerabend, Johannes M Mayrhofer, Tess B Oram, Ofer Yizhar, Jochen F Staiger, Carl CH Petersen | 2020 | Dataset for "Pathway-, layer- and cell-type-specific thalamic input to mouse barrel cortex" | https://doi.org/10.5281/zenodo.3560697 | Zenodo, 10.5281/zenodo.3560697 |

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
