## [Decision Letter]

**Acceptance summary:**

This paper provides a thorough, quantitative description of VPM and POm thalamocortical drive to different cell types and layers in S1 cortex. This type of data is essential for assembling an accurate circuit model of S1 function. Prior efforts to comprehensively quantify monosynaptic connectivity have focused on intracortical excitatory and inhibitory circuits, rather than the thalamocortical circuits studied here. While some features of VPM and POm input to S1 layers and cell types were known, this study provides the first full comprehensive mapping across excitatory and 3 inhibitory cell classes, across all layers. They use ChR2-assisted circuit mapping under monosynaptic conditions, which is the best high-throughput method available, and their dataset appears well-sampled both within and across individual experiments and carefully carried out.

**Decision letter after peer review:**

Thank you for submitting your article "Pathway-, layer-and cell-type-specific thalamic input to mouse barrel cortex" for consideration by *eLife*. Your article has been reviewed by three peer reviewers, including Sacha B Nelson as the Reviewing Editor and Reviewer #1, and the evaluation has been overseen by Eve Marder as the Senior Editor. The following individual involved in review of your submission has agreed to reveal their identity: Daniel E Feldman (Reviewer #2).

The reviewers have discussed the reviews with one another and the Reviewing Editor has drafted this decision to help you prepare a revised submission.

Essential revisions:

1) The reviewers agreed that the anatomical experiments in Supplementary Figure 1 were not sufficiently dispositive to be worth including.

2) The results and discussion concerning the short term plasticity for inputs to somatostatin positive interneurons should be curtailed. As suggested by one of the reviewers, "If they are arguing (based on Supplementary Figure 2) that no facilitation occurs, this is novel, [but] requires stronger proof. If they are arguing it does occur and these non-physiological pharmacological conditions conceal it, I don't follow why the figure should be included." It would perhaps be sufficient to state the fact that facilitation was not observed but that the conditions do not allow it to be measured physiologically.

3) The reviewers noted that a number of results had previously been reported in other studies and suggested that some statements, such as that "there has been no systematic quantification of VPM input across layers" was not warranted given e.g. data reported by Audette et al., 2018 and Petreanu et al., 2009. Additional care should be taken to make it clear to the reader which aspects are new.

---

## [Author Response]

Essential revisions:1) The reviewers agreed that the anatomical experiments in Supplementary Figure 1 were not sufficiently dispositive to be worth including.

We have removed Supplementary Figure 1 and the associated anatomical analysis.

2) The results and discussion concerning the short term plasticity for inputs to somatostatin positive interneurons should be curtailed. As suggested by one of the reviewers, "If they are arguing (based on Supplementary Figure 2) that no facilitation occurs, this is novel, [but] requires stronger proof. If they are arguing it does occur and these non-physiological pharmacological conditions conceal it, I don't follow why the figure should be included." It would perhaps be sufficient to state the fact that facilitation was not observed but that the conditions do not allow it to be measured physiologically.

We have removed Supplementary Figure 2 and now only briefly mention the lack of facilitation in the Results section.

3) The reviewers noted that a number of results had previously been reported in other studies and suggested that some statements, such as that "there has been no systematic quantification of VPM input across layers" was not warranted given e.g. data reported by Audette et al., 2018 and Petreanu et al., 2009. Additional care should be taken to make it clear to the reader which aspects are new.

We have rewritten several phrases to more accurately describe the advances brought about in this study. We have removed the phrase "there has been no systematic quantification of VPM input across layers". However, it should be noted that Audette et al., 2018, studied POm input (not VPM input), and although Petreanu et al., 2009, investigated both VPM and POm input, they did not study inhibitory neurons.